# Genome-wide analysis identifies Homothorax and Extradenticle as regulators of insulin in *Drosophila* Insulin-Producing cells

**Mattias Winant**[1], **Kurt Buhler**[1], **Jason Clements**[1], **Sofie De Groef**[1],
**Korneel Hens**[2], **Veerle Vulsteke**[1], **Patrick Callaerts**[1]*

**1** Laboratory of Behavioral and Developmental Genetics, Department of Human Genetics, KU Leuven—University of Leuven, Leuven, Belgium, **2** Centre for Functional Genomics, Department of Biological and Medical Sciences, Faculty of Health and Life Sciences, Oxford Brooks University, Headington Campus, Oxford, United Kingdom

☯ These authors contributed equally to this work.
* patrick.callaerts@kuleuven.be

**Data Availability Statement:** All relevant data are within the manuscript and its Supporting Information files. This includes the accession number for the RNAseq reads: European

## Abstract

*Drosophila* Insulin-Producing Cells (IPCs) are the main production site of the Drosophila Insulin-like peptides or dilps which have key roles in regulating growth, development, reproduction, lifespan and metabolism. To better understand the signalling pathways and transcriptional networks that are active in the IPCs we queried publicly available transcriptome data of over 180 highly inbred fly lines for *dilp* expression and used dilp expression as the input for a Genome-wide association study (GWAS). This resulted in the identification of variants in 125 genes that were associated with variation in *dilp* expression. The function of 57 of these genes in the IPCs was tested using an RNAi-based approach. We found that IPC-specific depletion of most genes resulted in differences in expression of one or more of the *dilps*. We then elaborated further on one of the candidate genes with the strongest effect on *dilp* expression, Homothorax, a transcription factor known for its role in eye development. We found that Homothorax and its binding partner Extradenticle are involved in regulating *dilp2*, *-3* and *-5* expression and that genetic depletion of both TFs shows phenotypes associated with reduced insulin signalling. Furthermore, we provide evidence that other transcription factors involved in eye development are also functional in the IPCs. In conclusion, we showed that this expression level-based GWAS approach identified genetic regulators implicated in IPC function and *dilp* expression.

## Author summary

Insulin signalling has a central and evolutionarily conserved role in many processes including growth, development, reproduction, lifespan, stress resistance and metabolic homeostasis. In the fruitfly Drosophila melanogaster insulin-producing cells in the brain are the main source of three insulin-like peptides, Dilp2, -3 and -5. How the production and secretion of these three insulin-like peptides are regulated remains incompletely understood. In the current study, genome-wide association studies were used to identify

Nucleotide Archive, Accession Number PRJEB50165.

**Funding:** P.C. was supported by FWO (Fonds Wetenschappelijk Onderzoek - Fund for Scientific Research Flanders) grants G065408.N10 and G078914N and by Onderzoeksraad, KU Leuven (Research Council KU Leuven) grant C14/17/099. K.H. acknowledges funding from the Biotechnology and Biological Sciences Research Council (grant BB/N00230X/1). K.B. was a predoctoral ('aspirant') fellow of the Fonds Wetenschappelijk Onderzoek (FWO). The funders had no role in study design, data collection and analysis, decision to publish, or preparation of the manuscript.

**Competing interests:** The authors have declared that no competing interests exist.

50 novel regulators of Dilp2, -3 and -5. We show that one of the top candidate regulators, Homothorax, is an important regulator of *dilp2*, *-3* and *–5* expression in the IPCs and is necessary for normal systemic insulin signalling and regulates adult size and developmental timing. We also show that the Hth interactor Extradenticle (Exd) is equally required in the adult but not in the larval IPCs. Finally, we show that most genes of the so-called retinal determination gene network are expressed in the IPCs and regulate normal *dilp2 and -5* expression. Together, these results identify further regulatory levels active in the IPCs and implicate a reshuffled version of a previously identified gene regulatory network therein.

## Introduction

Insulin/IGF signalling (IIS) plays a key role during growth and development but also in regulating reproduction, lifespan, stress resistance and metabolic homeostasis [1–4]. This central role is conserved across metazoans and beyond [5]. Eight Dilps (*Drosophila* Insulin-like peptides) have been described. Dilp1-6 are predicted to bind a single *Drosophila* Insulin Receptor (InR) that results in the activation of the conserved insulin signalling pathway [6]. Dilp7, produced by a group of sexually dimorphic neurons in the ventral nerve cord, acts on the receptor LGR4, and Dilp8, produced in the imaginal discs, binds the receptor LGR3 [1,7–10]. Dilp1-3 and 5 are produced in the brain by 14 larval and 16 adult insulin-producing cells (IPCs) [11]. Dilp1 is expressed only after pupation and regulates lifespan and metabolism epistatic to Dilp2 [12–14]. The function of Dilp4 is unknown and it is expressed only in the embryonic and larval midgut [1]. Dilp6 is more structurally homologous to mammalian Insulin-like growth factors and is produced in the fat body and glia. It plays an important role in the non-autonomous regulation of the IPCs [13,15]. Of all Dilps, those most homologous to mammalian insulin and also best studied are Dilp2, -3 and -5, produced in and secreted by the IPCs [16]. In adults, Dilp2 seems to be exclusively expressed in the IPCs, while Dilp3 and Dilp5 are additionally expressed in midgut muscle and in the principal cells of the Malphigian tubules in adults, respectively [17,18].

The Dilps produced by the IPCs have been shown to exhibit independent roles while also being partially redundant in their function. Δ*dilp2* mutants, for example, show increased *dilp3* and *dilp5* levels as a compensatory effect. Nonetheless, Δ*dilp2* mutants still have elevated haemolymph trehalose levels not seen in any other single *dilp* mutant. In addition, loss of *dilp2* has a prominent effect on growth, that is, however, further exacerbated by additional loss of *dilp3* and *dilp5* [2,19,20]. Additional evidence for this independent regulation comes from the numerous non-autonomous signals deriving from other tissues that have been shown to regulate the expression or secretion of only a subset of Dilps [21]. Some of these signals are larval or adult-specific. The fat body plays a key role in this as it relays the nutritional status of the organism to the IPCs to adjust systemic IIS levels [22]. Low nutritional protein levels, for example, induce the release of Eiger from the fat body into the haemolymph. Eiger binds to and activates its receptor Grindelwald on the IPCs to inhibit *dilp2* and *dilp5* expression but not *dilp3* [23]. Neuronal input differentially affects *dilp* expression. Knockdown of the octopamine receptor OAMB yielded increased *dilp3* expression while leaving *dilp2* and *dilp5* levels unaltered, while knockdown of the 5-HT1a results in increased *dilp2* and *dilp5* and unaltered *dilp3* [24]. The autonomous factors acting in the IPCs that mediate this independent regulation remain poorly understood.

To identify novel cell-autonomous regulators of IPC *dilp* expression we use the *Drosophila* Genome Reference Panel (DGRP) consisting of more than 200 highly inbred fly lines whose genome has been sequenced. The DGRP has been used effectively to identify genes that determine many quantitative phenotypes such as starvation resistance [25], aggressive behavior [26], mushroom body morphology [27], cocaine consumption [28] and lifespan [29]. While previous studies relied on quantitative behavioural or physiological parameters, we used the variation in *dilp2*, *-3* and *-5* mRNA levels of 183 (DGRP) [25,30] fly lines to identify genetic variants that correlate with *dilp2*, *-3* and *-5* expression levels. Genes associated with these variants were validated for a role in determining *dilp2*, *-3* and *-5* levels in the IPCs with an RNAi-based approach. We then elaborated on the function of one of the strongest candidate genes, the homeobox transcription factor Homothorax (Hth). We show that it is an important regulator of *dilp2*, *-3* and *–5* expression in the IPCs and is necessary for normal systemic insulin signalling and regulates adult size and developmental timing. We then show that the Hth interactor Extradenticle (Exd) is equally required in the adult but not in the larval IPCs. Based on the previously established roles of Ey, Dac and now Hth/Exd in regulating both IPC and eye field development we then hypothesized that other members of the Retinal Determination Gene Network (RDGN) could also be active in the IPCs [31–33]. We show that most RDGN genes are expressed in the IPCs and that they are mostly necessary for normal *dilp2 and -5* expression. These results point to the existence of a reshuffled RDGN being active in the IPC to control insulin expression.

## Results

### GWAS for *dilp2,3* and *5* expression levels reveals sexually dimorphic putative regulators

We collected whole body expression data for *dilp2*, *-3 and -5* of 183 DGRP lines [30] (Fig 1A and S1 Table). Considerable interline variation is observed and significantly higher read counts were detected in males compared to females for the three *dilps*. We then ran a GWA analysis for each of the *dilps* and identified a total of 271 variants that were associated with at least one *dilp* and in one sex (Fig 1B). These 271 variants correspond to 125 unique genes (Fig 1B). Based on RNA-seq data of the adult IPCs to ensure that genes were expressed in the adult stage (S2 Table) and RNAi line availability we assessed the role of 57 candidate genes (S3 Table) in regulating *dilp2–3* and *-5* expression in the IPCs using qPCR. This showed that all but 7 had significant effects on *dilp2*, *-3* or *-5* regulation in one or both sexes (Fig 1C).

Most genes that were identified in the GWAS were identified in only one sex suggesting a highly sexually dimorphic regulation of *dilp* expression (Fig 1B). However, when comparing changes in expression of the *dilps* upon knockdown of the 57 candidate genes in the IPCs we noticed that the *dilp* levels in both sexes showed significant correlation (Fig 2A, 2B and 2C). To better understand this apparent contradiction with the GWAS results, we classified per *dilp* each candidate gene as sexually dimorphic if RNAi mediated depletion in the IPCs resulted in a significant reduction or increase of *dilp* mRNA levels in one sex only, or if it had an opposite effect on *dilp* levels in both sexes. The latter possibility was only seen for *Ldsdh1* where knockdown resulted in increased *dilp2* levels in females and decreased *dilp2* levels in males (Fig 2A'). Overall, we found that out of 57 candidate genes, knockdown of 34 genes resulted in changes in *dilp2* expression of which 79% (27) were sexually dimorphic (Fig 2A'). Changes in expression of *dilp3* were seen for 34 out of 57 candidate genes with 76% sexually dimorphic (Fig 2B'). Finally, 30 out of 57 genes resulted in altered expression of *dilp5* with 30% being sexually dimorphic (Fig 2C'). Of note, the genes that result in changes in *dilp* expression upon knockdown are not the same for each *dilp*, consistent with the hypothesis that there is significant

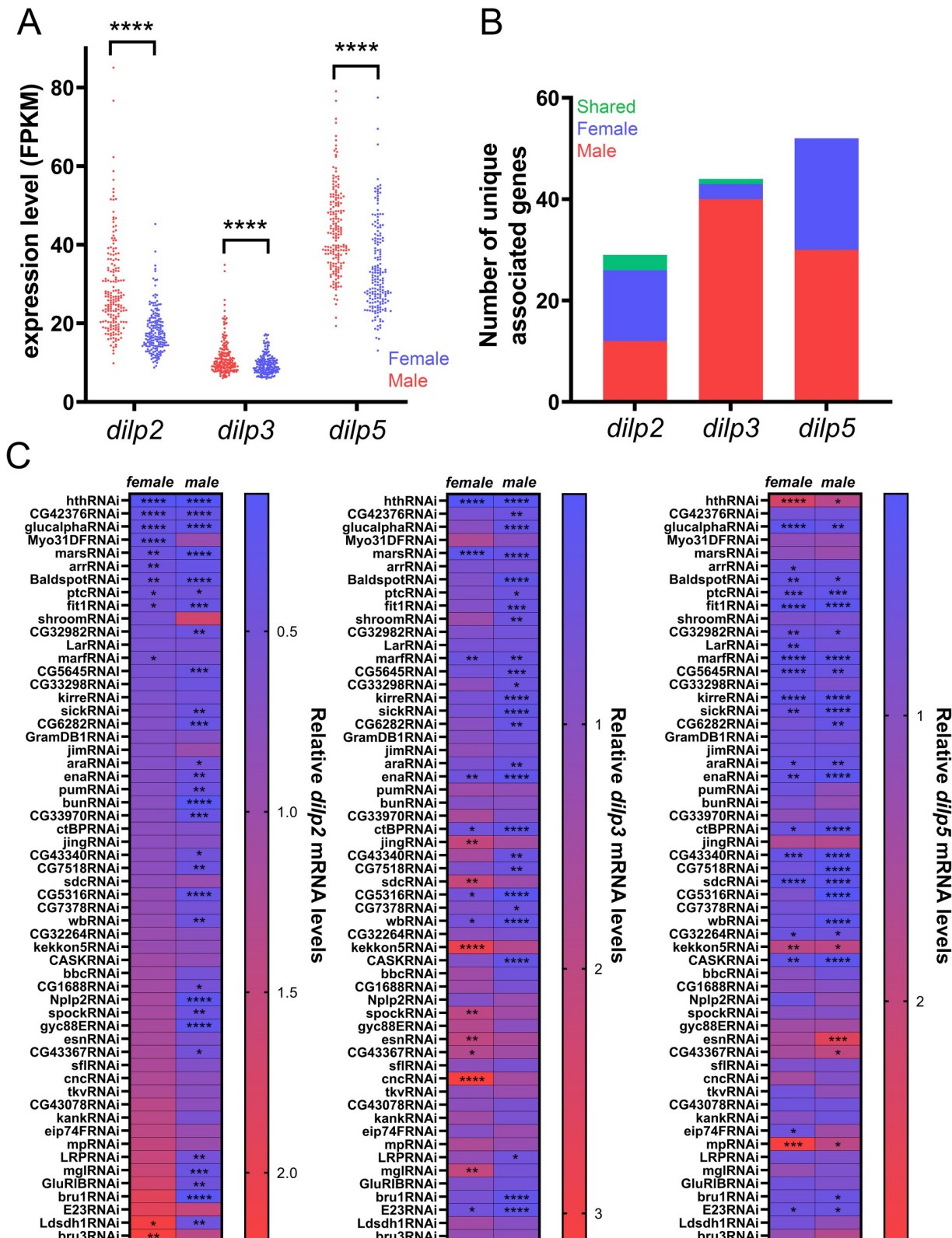

**Fig 1. DGRP Screen and validation.** (A) Mean *dilp2, -3 and dilp5* FPKM levels were collected from Huang et al.[30] and are significantly higher for males than females for *dilp2* (p < 0.0001, unpaired Welch's T-test, n = 183), *dilp3* (p < 0.0001, unpaired Welch's T-test, n = 182) and *dilp5* (p < 0.0001, student's T-test n = 182). (B) Number of unique genes associated for each dilp/sex (male dilp2: 12, female dilp2: 14, shared dilp2: 3,

male dilp3: 40, female dilp3: 3, shared dilp3: 2, male dilp5: 30, female dilp5: 22 and shared dilp5: 0). (C) Mean qPCR levels for *dilp2*, *-3* and *dilp5* upon RNAi-mediated knockdown using *Dilp2-GAL4$^R$;btlGAL80* (values are means of the relative mRNA quantity of n = 3 consisting of 10–15 fly heads each). Results of female *dilp2* levels were used to sort the RNAi lines (low to high). (* p < 0.05, ** p <0.01, *** p<0.001, **** p < 0.0001, Dunnett's multiple comparisons compared to an *mCherry$^{RNAi}$* control line, which itself had no effect on *dilp* expression (S1 Fig).

independent regulation of transcription for the *dilps*. In conclusion, we identify significant sexually dimorphic effects for regulation of *dilp2* and *dilp3* and less for *dilp5*.

## Homothorax knockdown reduces *dilp2* and *-3* mRNA expression and Dilp2, -3 and -5 protein levels

The strongest decline in both *dilp2* and *dilp3* levels in males and females was noted for the transcription factor Homothorax. IPC-specific downregulation resulted in significant downregulation of both *dilp2* and *dilp3* mRNA levels while *dilp5* mRNA levels were significantly increased (Fig 3D, 3F and 3H)). First, we validated the expression of Hth within the IPCs of both the L3 and adult stage by using antibodies directed against Hth (Fig 3A and 3B). We then elaborated on the effect of Hth on *dilp* expression by characterising the effect on Dilp protein levels by using specific antibodies against Dilp2, -3 and -5 [34]. When compared to sibling controls this revealed a strong reduction of Dilp2–3 and -5 immunoreactivity within the IPC somata (Fig 3C–3G'). Dilp2-GAL4 was used to knock down Hth levels. Since Dilp2-GAL4 is controlled by *dilp2* regulatory sequences, it is possible that Hth knockdown could affect GAL4 activity. To verify this, we quantified IPC GFP levels of *Dilp2-GAL4> nlsGFP;hth$^{RNAi}$* and found a reduction of GAL4 activity (S2A–S2C Fig). However, when we used a second GAL4 line (*OK107-GAL4*, inserted in *eyeless*, and not regulated by Hth (S3 Fig) [35]), the observed reductions in *dilp2* mRNA levels in both males and females were recapitulated showing that the effect of *hth* knockdown on *Dilp2-GAL4* activity has a negligible impact on the observed phenotypes (S2D and S2E Fig).

## Knockdown of the Hth co-factor, Extradenticle, induces a similar reduction in *dilp* expression and Dilp protein levels as observed upon Hth depletion

During development, Hth physically interacts with the homeobox transcription factor Exd. Exd forms a heterodimer with Hth and is necessary for nuclear localization of Exd [36]. An α-Exd antibody detects protein expression in both L3 and adult IPCs (Fig 4A and 4B). We therefore hypothesized that Exd, together with its co-factor Hth, regulates *dilp* expression in the IPCs. IPC-specific knockdown of *exd* strongly reduced expression of *dilp2 and -3* while significantly increasing *dilp5* mRNA levels (Fig 4D, 4F and 4H). Dilp protein levels were also reduced in the adult stage (Fig 4C–4G'). We then asked whether the known dependence of Exd on Hth for its nuclear localization is also at play in the IPCs. RNAi experiments in the IPCs showed that the nuclear localization of Exd in the IPCs is dependent on normal Hth expression in the IPCs in both larval and adult IPCs but the converse is not true (S4A–S4D Fig). Hth and Exd are thus equally required for normal *dilp* expression and Dilp protein levels.

## Hth and Exd both regulate adult systemic insulin signalling but Exd is dispensable for larval stage-associated phenotypes

IIS and *dilp2* in particular play key roles in the regulation of growth and development during the larval stage [2,16]. We therefore hypothesized that IPC-specific knockdown of *hth* and *exd* knockdown would result in decreased adult size [2,16]. *hth* but not *exd* knockdown results in decreased adult weight, head width and wing area (Fig 5A, 5B and 5C). Developmental timing,

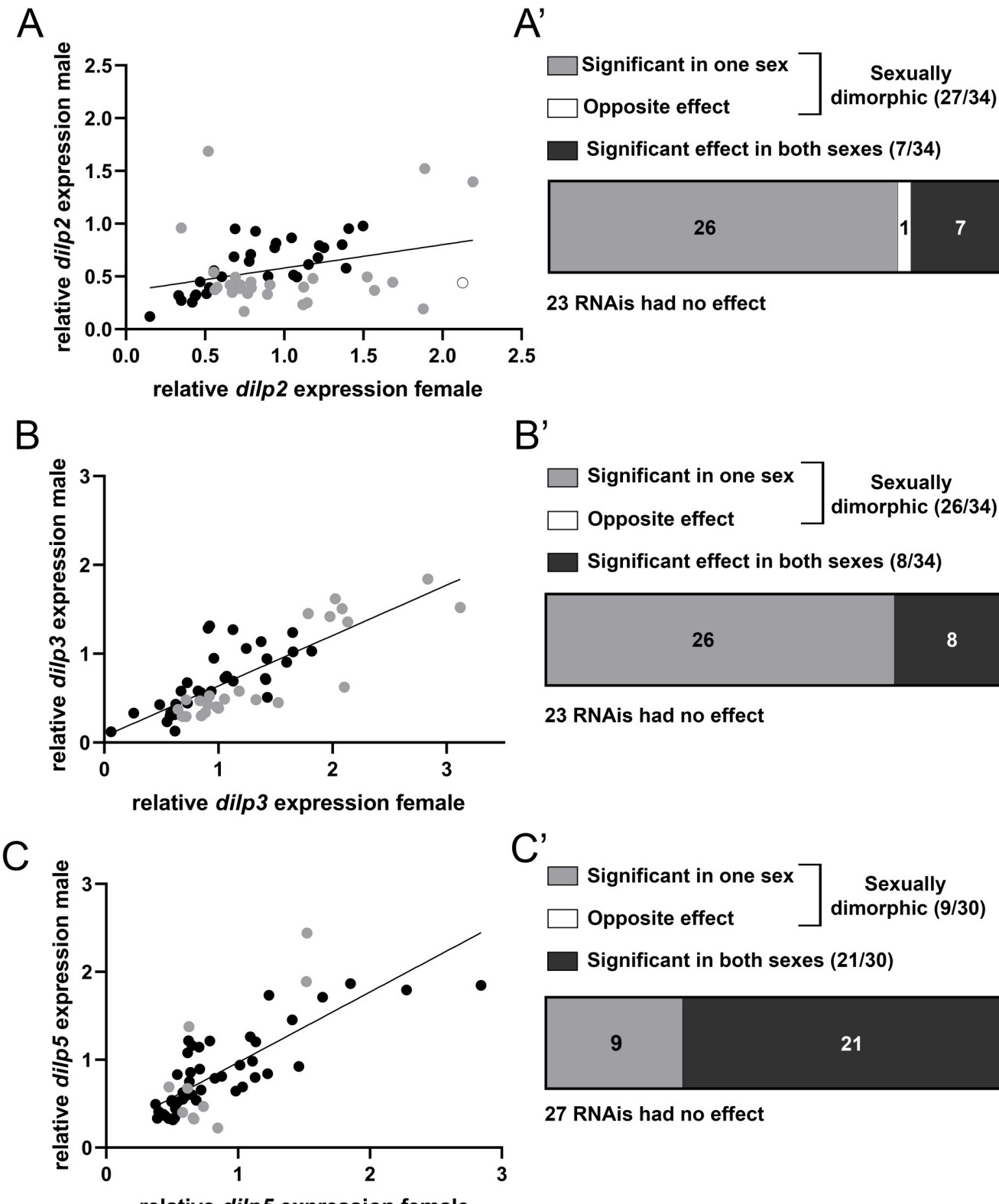

**Fig 2. Knockdown of candidate genes in the IPCs produces correlated effects on *dilp* expression between the sexes.** (A) The mean relative quantity of *dilp2* upon RNAi-mediated knockdown in the IPCs between female and male flies is mildly correlated (Spearman r = 0.3467, p = 0.0081). (A') Out of 57 tested genes, 27 had sexually dimorphic effects on *dilp2* expression, one (*Ldsdh1*) having opposite effects on *dilp2* expression. 30 genes had no effect (23) or

the same effect (7) on *dilp2* levels. (B) Relative *dilp3* levels between the sexes are highly correlated. (Spearman r = 0.7955, p < 0.0001) (B') 26 genes had a sexually dimorphic effect on *dilp3* expression, 31 had no effect (23) or the same effect (8). Relative quantity levels of *dilp5* are correlated (Spearman r = 0.7699, p < 0.0001). (C') 9 genes had a sexually dimorphic effect on *dilp5* levels while 48 genes had no effect (27) or the same effect (21) on *dilp5* expression.

as measured by time until eclosion, is also increased only upon *hth* depletion (Fig 5D). Adult fat content was increased in *hth* knockdown only (Fig 5E). Oxidative stress resistance, induced by the addition of paraquat in the food, was increased in both *exd* and *hth* knockdown (Fig 5F). Lastly, female fecundity is reduced in both *exd* and *hth* knockdown flies (Fig 5G). Based on the reduction in fecundity and the increased lifespan under oxidative stress conditions we conclude that Hth and Exd are essential for normal systemic insulin signalling in adults. The size and developmental timing phenotypes, on the other hand, show that Exd is dispensable for normal larval systemic insulin signalling while Hth is necessary during the larval stage. We next wondered if overexpressing Hth and Exd in the IPCs is sufficient to increase *dilp* expression. No significant effects on *dilp2*, *-3* and *-5* mRNA levels were observed (S5A Fig). Overexpressing Hth or Exd also left adult weight unaffected (S5B Fig). We next checked if Hth and Exd are sensitive to different nutritional conditions or if they function independently of nutritional status. Flies were placed on starvation, sucrose only or protein rich media for 72h and the presence of Hth/Exd in IPC nuclei was assessed (S6A–S6C Fig). This revealed that there were no appreciable differences in nuclear localization of Hth/Exd. Moreover, flies depleted of *hth* or *exd* in the IPCs in these three different nutritional conditions show similar reductions in *dilp2* and *-3* mRNA levels and increased *dilp5* levels as observed for flies on control food (S6D–S6G Fig). This shows that Hth-Exd play a role in the constitutive regulation of *dilp* mRNA levels and are not influenced by the nutritional status of the fly. Furthermore, the presence of multiple Hth binding sites in promotor-enhancer regions of *dilp2*, *-3* and *-5* genes suggests that they may be directly regulated by Hth (S7A–S7C Fig).

## Hth and Exd have different effects on dilp regulation in larvae

To address the different effects of Hth and Exd adult size, we looked at dilp levels in the larval stage. This revealed that already at the late L2 stage (68h After Egg Laying (AEL)) Dilp2 protein levels are reduced by *hth* depletion but not upon *exd* depletion (S8A and S8B Fig). Later during development, at mid L3 (92h AEL), Dilp2, *-3* and *-5* protein levels are reduced in both *hth* and *exd* knockdown (S9A–S9F Fig). Later in larval development at 100h AEL, prior to wandering, immunohistochemistry revealed that Dilp2 protein levels are reduced in both Hth and Exd depletion (Fig 6A and 6C), while Dilp5 levels are unaffected by *exd*$^{RNAi}$ but reduced by *hth*$^{RNAi}$ (Fig 6B and 6D). Furthermore, mRNA levels of *dilp2* and *-5* are reduced only for *Hth*$^{RNAi}$ but not *Exd*$^{RNAi}$ at 100h AEL (Fig 6E and 6F). Dilp2, the main regulator of growth, is thus severely reduced at all checked timepoints upon Hth depletion while the effect of Exd depletion on Dilp2 is more subtle where Dilp2 protein levels are reduced at late L3, but mRNA levels are not significantly decreased which is suggestive of increased Dilp2 secretion. In conclusion, we show that Hth is necessary for Dilp regulation from at least late L2 onwards while the larval role of Exd is more complex.

## Retinal Determination Gene Network is active in the IPCs to regulate *dilp* expression

In the mammalian pancreas, the Hth/Exd homologs Meis1/PBX1/2 function upstream of PAX6/Dach1/2 where they are required to induce *pax6* expression [37,38]. We thus hypothesized that Hth would have a similar regulatory relationship with Ey in the IPCs. Depletion of

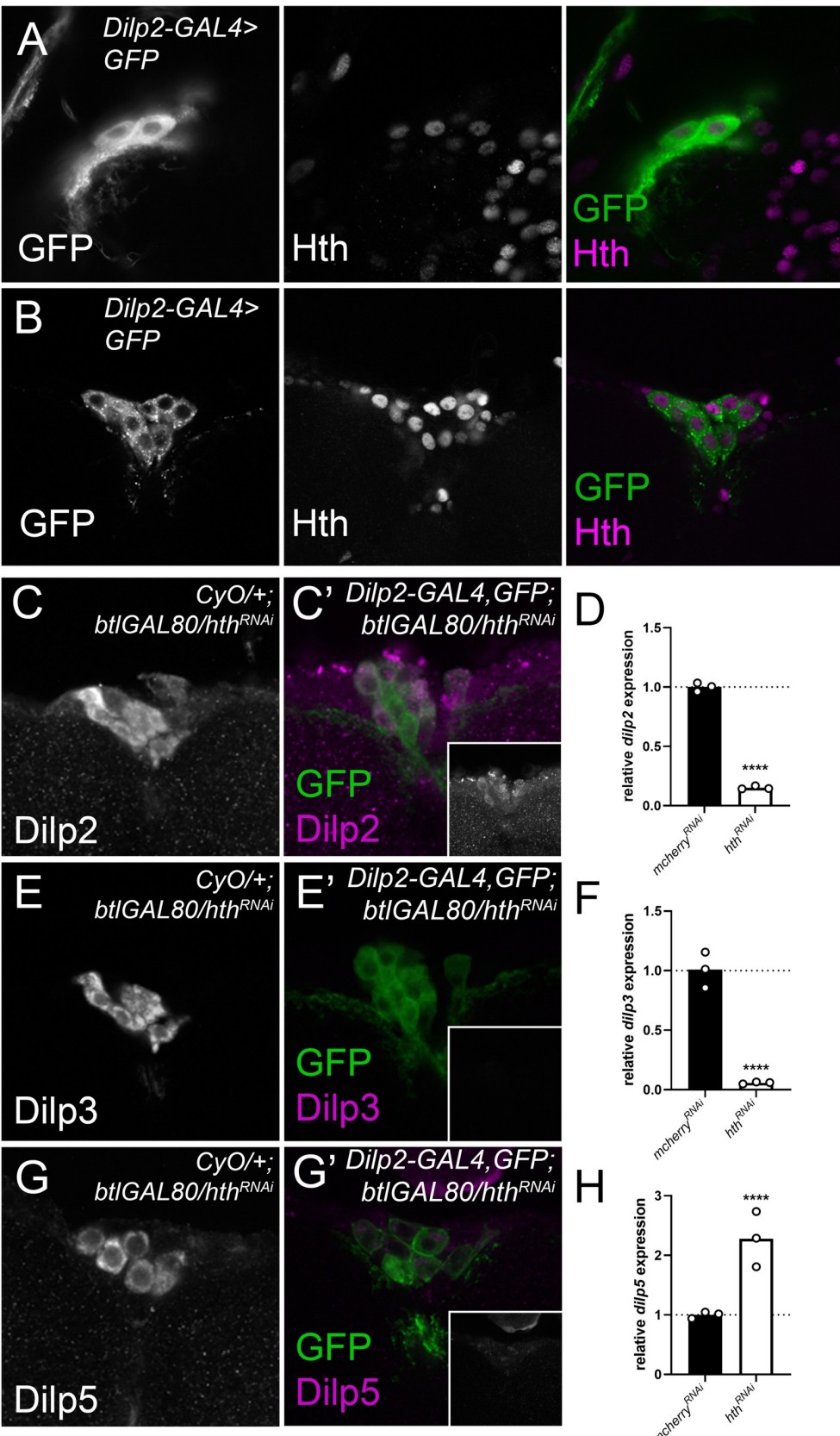

**Fig 3. Hth is expressed in larval and adult IPCs and is necessary for normal Dilp protein levels.** Antibody staining detects Hth in L3 (A) and adult (B) nuclei in IPCs that are marked by CD8-GFP expression. Knockdown of *hth* reduces Dilp2 (C-C'), Dilp3 (E-E') and Dilp5 (G-G') protein expression in the IPCs compared to balanced age-controlled siblings (left) reared in identical environmental conditions. Insets (C', E', G') show Dilp2, -3 and -5 only (grayscale). Brains from knockdown and balanced siblings were dissected and stained in the same tube. Expression of CD8-GFP was used to distinguish control and test genotypes, and to label the IPCs. Images were acquired with identical confocal settings and representative examples are shown. Expression of *dilp2* (D) and *dilp3* (F) is significantly reduced and *dilp5* (H) is significantly increased upon Hth depletion in the IPCs (*dilp2*, *-3* and *-5* qPCR results are added to facilitate comparison with Dilp protein results, and correspond to the Hth-related results shown in Fig 1C).

*hth* in the IPCs did not result in differences in Ey levels in the IPCs, while Exd expression was no longer detectable in the IPC nuclei, showing that Hth is required for normal Exd localization and/or expression, but not for normal Ey expression (S3 and S4 Figs).

Hth/Exd are functional in the retinal determination gene network (RDGN), necessary for eye development [33]. We and others had already described a role for other RDGN members: Eyeless (Ey) and Dachshund (Dac) in regulating the IPCs. Therefore, we hypothesized that a version of this regulatory network controls *dilp* expression in the IPCs [31,32]. To test this, we first queried the transcriptome of both larval and adult IPCs (Cao et al [39], and this study) for expression of 11 different RDGN members. Of these 11 TF, all were expressed to some degree in the adult IPCs, with *hth*, *gro*, *optix*, *dac*, *eya*, *ey* and *dan* showing expression in both stages. (Fig 7A). These results were validated with antibody staining, transgenic GAL4 reporter lines and tagged versions where available (Fig 7B–7W). Only Eya showed no detectable expression in the larval or adult IPCs but the overall staining quality for Eya was poor (Fig 7P and 7Q) and while RNA-seq results for the larval stage suggested that only a subset of RDGN TF were expressed in L3 IPCs, all but Eya are clearly detectable at the protein level using immunohistochemical reagents while Eyg, Toe, Eya, Dan, and So are not detected in the adult stage (Fig 7M, 7O, 7Q, 7U and 7W). This shows that most RDGN transcription factors are expressed in the IPCs. To establish a role for these transcription factors in the IPCs, we genetically depleted these TFs in the IPCs using *Dilp2-GAL4*[215-1-1-1] and their respective RNAi lines [40]. Except for *dac* and *toe*, knockdown of these TFs resulted in a significant reduction in *dilp2* and *dilp5* expression compared to D*ilp2-GAL4*[215-1-1-1] flies driving a control RNAi under identical environmental conditions (Fig 5X, 5Y and 5Z). Knockdown of *dac* reduced *dilp5*, and increased *dilp2* levels, while knockdown of *toe* only reduced *dilp5* expression. Furthermore, only the depletion of *tsh* and *ey* affected *dilp3* levels (Fig 5S). Since reduced *dilp2* and *-5* levels could be due to a reduction in the number of IPCs, we verified this and found that the number of IPCs are unaltered (S10 Fig). Overall, these results suggest that the RDGN is re-used in the regulation of IIS in the IPCs.

In summary, these data show that a repurposed RDGN is used in the IPCs. Knockdown of *hth* and *exd* affects all three *dilps*. On the other hand, most RDGN TFs regulate either *dilp2* and *dilp5*, while *toe* regulates only *dilp5* and *tsh* regulates all three. Our data provide an interesting example of how regulatory networks are re-used in other developmental contexts to achieve different transcriptional outcomes.

## Discussion

The IPCs are key regulators of growth and metabolism in the fly but the regulatory networks governing their development and function remain poorly understood. In this study, we use the variation in *dilp* expression across DGRP lines to identify genes that are necessary for normal IPC-autonomous *dilp* regulation. There was very little overlap between the sexes for the identified candidate genes (3 genes for *dilp2*, 2 genes for *dilp3* and none for *dilp5*). Furthermore, only one gene was shared between the dilps (*Glucalpha* for *dilp3* and *dilp5* in females).

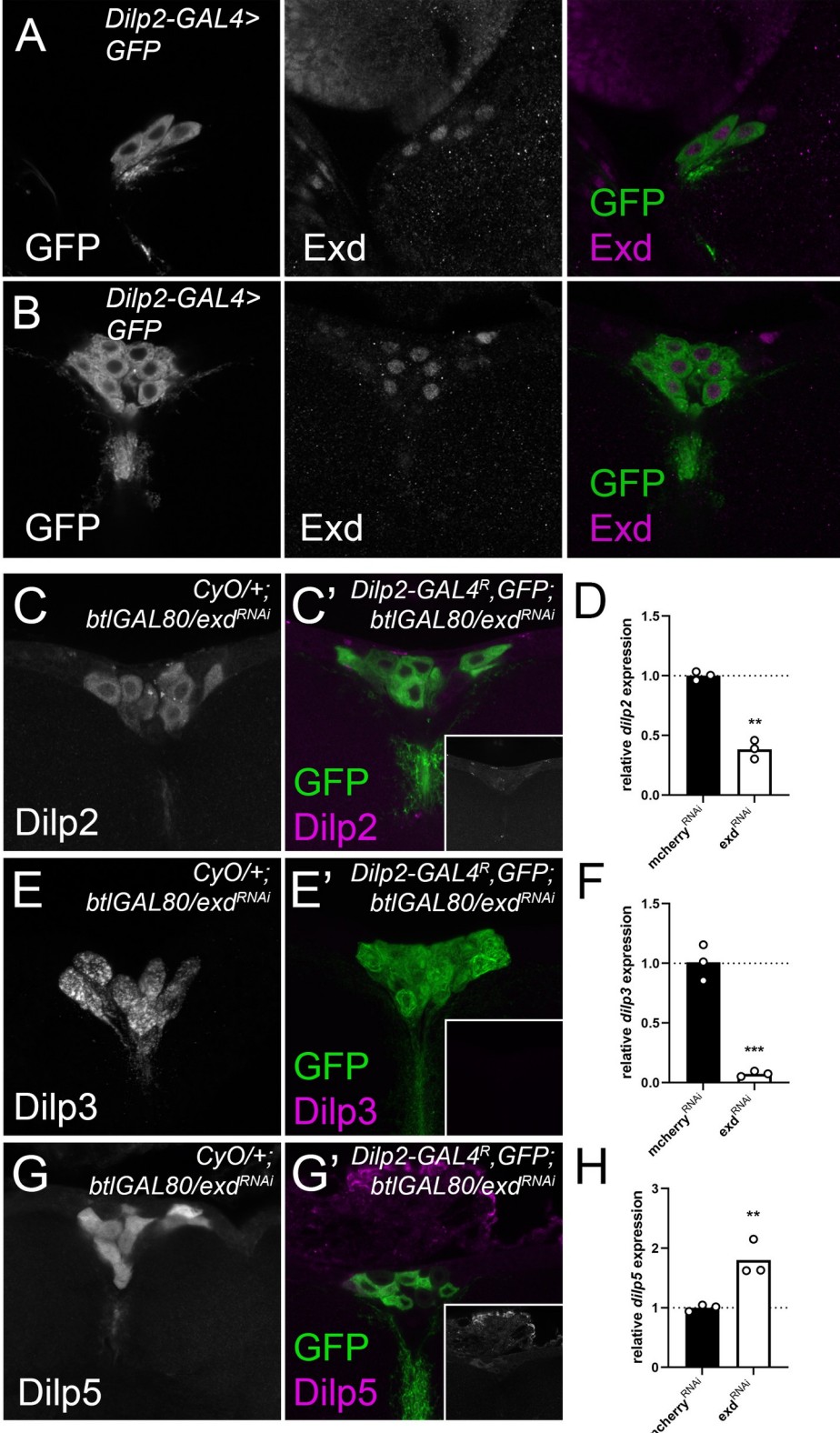

**Fig 4. The Hth interactor Exd, is expressed in the IPCs and is equally required for *dilp* expression.** (A-B) Antibodies against Exd detect expression in L3 and adult IPC nuclei, marked by *dilp2-GAL4^R* driven *UAS-CD8GFP expression*. (C-C') Dilp2, (E-E') Dilp3 and Dilp5 (G-G') protein levels are reduced in adult IPCs upon Exd depletion in

the IPCs compared to balanced age-controlled siblings. Insets (C', E', G') show Dilp2, -3 and -5 only (grayscale). Brains from knockdown and balanced siblings were dissected and stained in the same tube. Expression of CD8-GFP was used to distinguish control and test genotypes, and to label the IPCs. Knockdown of *exd* in the IPCs reduced expression of (D) *dilp2* and (E) *dilp3* significantly while *dilp5* is upregulated. Flies were reared at controlled densities. RNA was collected from 10–15 female fly heads. After normalization, *dilp* expression was normalized to a control *dilp2-GAL4R; btlGAL80* driving *UAS-mCherry*[RNAi].

The results from our RNAi based approach to validate the role of these putative regulators revealed that depletion of most genes had sexually dimorphic effects, yet also that the effects on *dilp* expression between the sexes were correlated (Figs 1C and 2A–2C'). Additionally, genes implicated in the regulation of one *dilp* often also had effects on the other dilps. Variants in Hth, for example, were identified to correlate with *dilp5* levels in males but not females. Depletion of Hth in the IPCs produced a pronounced effect on *dilp5* levels and additionally resulted in a strong reduction of both *dilp2* and *dilp3* levels in both sexes (Fig 1C). Several possible explanations for this apparent discrepancy exist. One explanation is that our RNAi-based approach might have obscured the effect of the specific variants which might have sex-specific and/or *dilp*-specific effects. The compensatory effects previously noted in *dilp* regulation might additionally add another layer of complexity onto our results [2,41]. Furthermore, the identified genes might play an additional role in the non-autonomous regulation of *dilp* expression by acting in other tissues to regulate *dilp* expression [21]. Lastly, it is possible that identified genes are associated with multiple *dilps* and in both sexes but that they did not reach the significance threshold due to insufficient statistical power.

Our approach revealed that Hth is required in the adult IPCs to regulate *dilp* expression (Fig 1C). IPC-specific depletion of Hth resulted in almost complete absence of *dilp2* and *dilp3* mRNA in both sexes (~10% of control) while doubling *dilp5* mRNA levels. Immunohistochemistry confirmed the strong reduction of Dilp2 and Dilp3 on the protein level but an increase on Dilp5 protein levels was not seen. The latter could be due to increased Dilp5 secretion or a failure to produce properly matured Dilp5 protein (Fig 3C, 3D and 3E). The increase in *dilp5* mRNA levels is reminiscent of what is seen in *Δdilp2-3* deletion mutants which show upregulated *dilp5* mRNA levels, presumably to partially compensate for loss of *dilp2* and *-3* [2].

Further phenotypic characterization revealed that Hth but not Exd is essential for both normal development and growth at the larval stage (Fig 5A, 5B, 5C and 5D). As both larval growth and developmental timing are disturbed upon *hth* depletion, this suggests that IIS levels are reduced before and after attainment of critical weight [42,43]. Exd seems dispensable for both growth and developmental timing although IPC Dilp2, -3 and 5 protein levels are reduced at mid L3. By late L3, however, *dilp2* and *-5* mRNA levels are not significantly altered (Fig 6E and 6F) and Dilp2 protein levels are reduced in the IPCs (Fig 6D). This suggests that Dilp2 secretion might be increased thereby minimizing growth defects. Regulation of *dilp2*, *-3* and *–5* in larvae by Hth thus seems to be partially independent of Exd and depends on the specific timepoint during larval development. During larval development several peaks in the titre of the steroid hormone, Ecdysone, mark the transitions between larval stages and more subtly in L3, marking critical weight and the transition towards wandering [43]. Recent work from our lab showed that the larval IPCs are Ecdysone responsive [34]. This raises the possibility that the interaction between Hth-Exd might itself be Ecdysone sensitive, explaining their effects in larval *dilp* regulation.

Further analysis revealed that Hth and Exd had similar effects on regulating *dilp* expression in adults. Depleting Exd and Hth in the IPCs decreased female fecundity and increased oxidative stress resistance consistent with a reduction in systemic insulin signalling (Fig 5F and 5G).

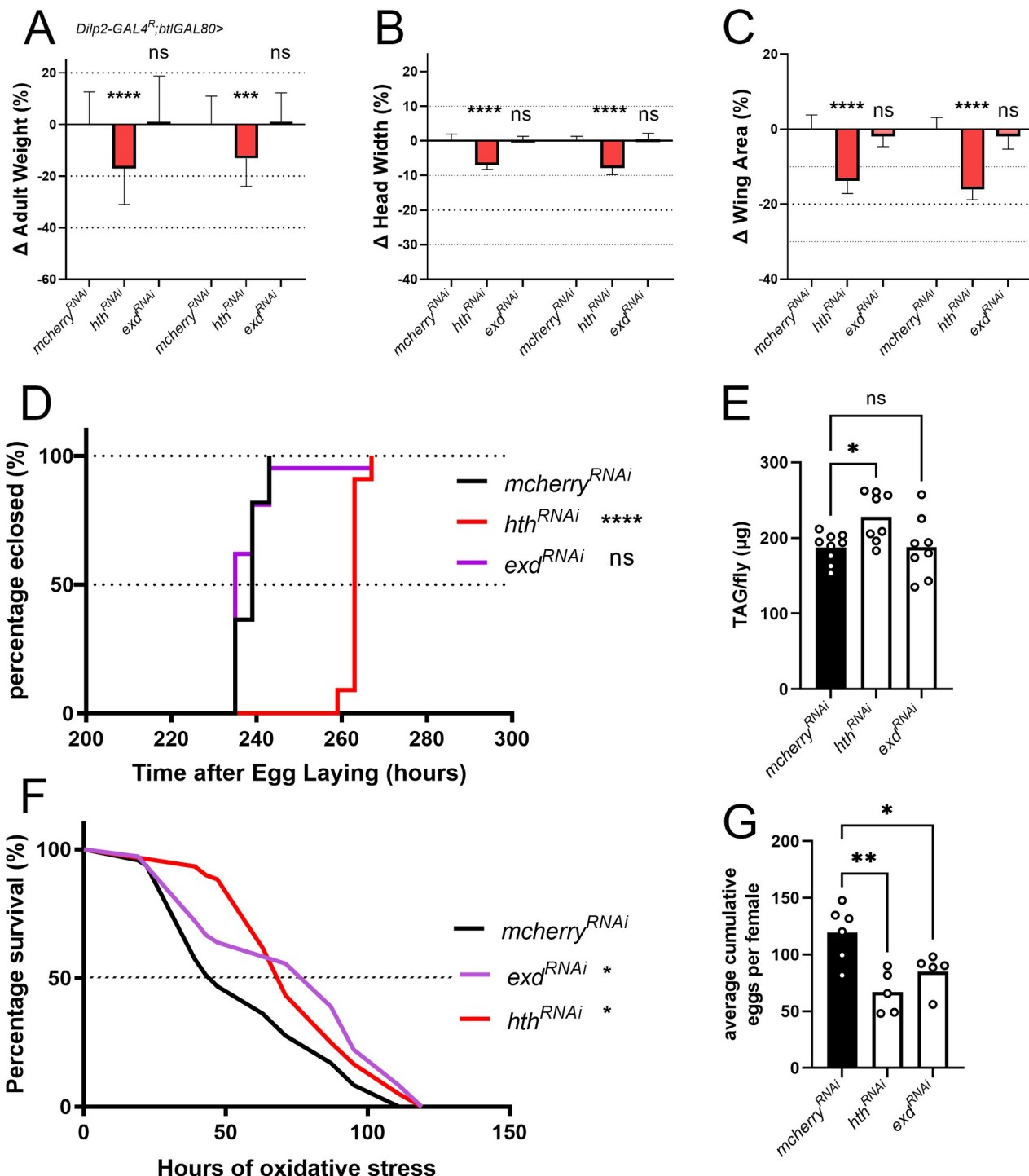

**Fig 5. Depletion of Hth and Exd in the IPCs results in phenotypes associated with reduced systemic insulin signalling.** (A-C) Knockdown of *hth* but not of *exd* results in decreased adult weight, head width and wing area (one-way ANOVA with post-hoc Dunnett's test comparing to *mCherry*[RNAi] controls; ns, not significant, *** p<0.001, **** p < 0.0001). Embryos were collected on yeasted apple-juice plates for 3 hours and allowed to develop to early L1 larvae when they were transferred to food vials at a fixed density (20 larvae per vial). Flies were anaesthetized with chloroform 4–5 days after eclosion for measurements. (D) Knockdown of *hth* in the IPCs but not of *exd* results in a developmental delay (one-way ANOVA with post-hoc Dunnett's test comparing to *mCherry*[RNAi] controls; ns, not significant **** p < 0.0001). (E) Total TAG of female flies is increased upon Hth knockdown but not in case of Exd knockdown (one-way ANOVA with post-hoc Dunnett's test comparing to *mCherry*[RNAi] controls; ns, not significant * p < 0.05). (F) Oxidative stress resistance is increased in flies expressing *hth*[RNAi] and *exd*[RNAi]. Flies were reared on normal food and aged to 3 days old and placed at controlled densities on food containing 20mM paraquat. Dead flies were scored regularly (* p

<0.05,** p<0.01, ns, not significant, log-rank test with Bonferroni correction for multiple testing). (G) IPC-specific depletion of hth or exd resulted in decreased female fecundity. Every data point is the average (4–5 females per vial) cumulative number of eggs layed for 14 days (one-way ANOVA with post-hoc Dunnett's test comparing to *mCherry*^RNAi controls * p <0.05,** p<0.01).

By contrast, total TAG in adult flies is increased only when *hth* is knocked down. Recent work showed that *dilp2* knockdown in the IPCs in the larval stage only is sufficient to increase total TAG levels in adult flies [44]. Consistently, we find that Exd knockdown does not significantly alter *dilp2* mRNA levels in late larvae (Fig 6E).

In the mammalian pancreas the fly homolog of Ey, Pax6, is required to establish the cellular fate of endocrine cell types and to promote expression of their respective hormone gene products. To do this, Pax6 cooperates with Dach1/2, which is also observed in the Ey/Dac regulation of *dilp5* in *Drosophila* [32]. Upstream of Pax6, the Meis gene family members Meis1/2/ Prep1/2 and Pbx1/2, which are homologs of *Drosophila* Hth and Exd, are required to promote pancreatic *pax6* expression [37]. Depletion of Hth in the IPCs, however, leaves Ey expression unaffected while resulting in the absence of Exd in IPC nuclei (S5 and S7 Figs). This shows that while the TFs are identical between flies and humans, their regulatory relationships are not. Similarly, differences in regulatory relationships are also seen during eye development where Meis1/2 regulate *pax6* in the mouse lens. In *Drosophila*, however, Hth has no direct regulatory relationship with *ey* or its paralog *toy* [45,46]. Key TFs are thus conserved in *Drosophila*/mammalian eye development and IPC/pancreas development while the regulatory relationships between the TFs have diverged during evolution.

We further show that the genes of the RDGN are active in the IPCs to regulate insulin signalling, mostly via control of *dilp2* and *dilp5* expression. Of all tested TFs only Ey and Hth/Exd are necessary for promoting *dilp3* expression while Tsh depletion resulted in an upregulation of *dilp3* levels (Fig 7Y). Other RDGN TFs were not required to regulate *dilp3*, and most regulated *dilp2* and *dilp5*. In literature, co-regulation between *dilp2* and *-5* is commonly reported, despite them having distinct functions [19]. However, given the complex regulatory relationships between the RDGN TFs in the developing eye, it is surprising to see that, in general, their regulatory relationship is largely positive, promoting expression of these two *dilps* (Fig 8A and 8B). One explanation is that since a couple of the RDGN TFs are already known to be expressed at the neuroblast stage [47], depleting these TFs during development might result in a failure to properly establish IPC cell fate. Additionally, a more complex regulatory architecture may exist that is relevant in other physiological contexts to which the *dilps* are sensitive, such as in changing dietary conditions, aging, or oxidative stress [48]. Of note are the opposing effects of Dac and Ey on *dilp2* expression, while still physically interacting to promote *dilp5*. This implies the presence of *dilp*-specific regulatory relationships. We do, however, not provide evidence for the direct regulation of the *dilps*, but we hypothesize, based on the strong effects of Hth/Exd on *dilp* expression and on the presence of Hth binding sites within *dilp2*, *-3* and *-5* enhancers that they may directly regulate the *dilps*, particularly *dilp2* and *dilp3*, while others may contribute via indirect interaction or regulation of additional factors (Fig 8B).

Interestingly, a study by Okamoto *et al.* [32], which originally identified Dac as a co-regulator of *dilp5* together with Ey, also knocked down a number of other RDGN TFs (*optix*, *eya*, *eyg*, *tio*, *so*, *toy*). They find small and often not significant effects on *dilp* levels when compared to our results. We speculate that differences between our results and theirs lay in the age of the flies, the different *dilp2*-GAL4 line that was used or the differences in dietary conditions. Another study based on type 2 diabetes risk loci in humans by Peiris *et al.* [49], showed that IPC-specific knockdown of *Optix* resulted in decreased circulating Dilp2 levels. Our results suggest that this reduction in Dilp2 release might be indirect due to the reduction of *dilp2* mRNA levels upon *Optix* depletion.

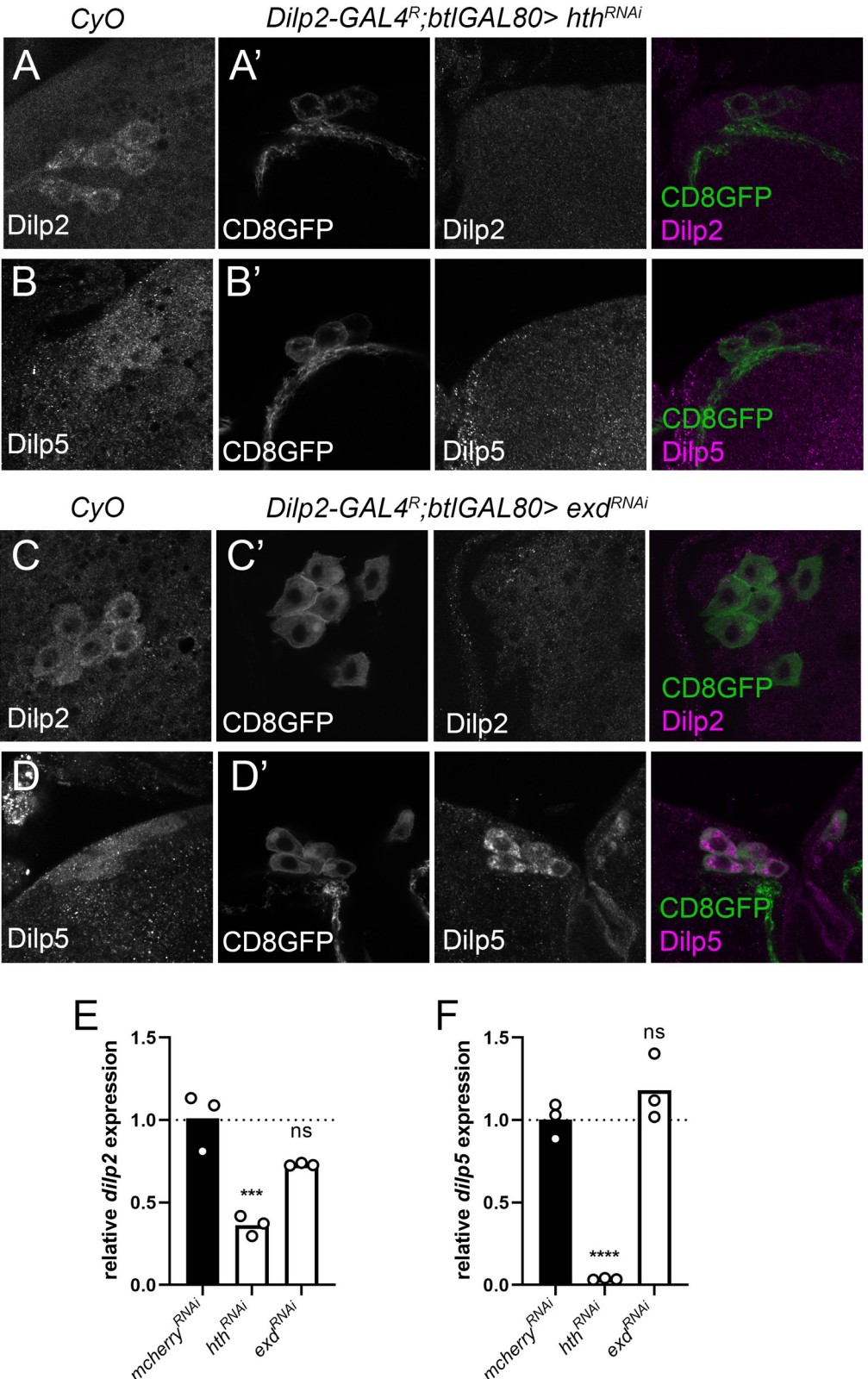

**Fig 6. Hth and Exd have differing effects on Dilp protein and mRNA levels in late L3.** (A-A', B-B') representative images showing reduced Dilp2 and Dilp5 protein levels in late L3 (100h after egg laying) larvae genetically depleted of *hth* compared to balanced sibling controls (CyO, left panels). (C-C', D-D') IPC-specific *exd* knockdown results in

decreased Dilp2 and unaltered Dilp5 protein levels in IPC somata. (E) *dilp2* mRNA levels are significantly reduced upon *hth* depletion and not significantly reduced upon *exd* depletion. (F) *dilp5* mRNA levels are severely reduced *hth* knockdown larvae and are unaltered in *exd* knockdown. (ns, not significant, ***p<0.001, ****p<0.0001), one-way ANOVA with post-hoc Dunnett's test to correct for multiple comparisons).

To conclude, our approach identified many novel regulators of *dilp* expression with a prominent role for Hth/Exd in regulating *dilp* expression in both sexes. In addition, we show that the RDGN is repurposed to regulate the development and function of the IPCs. The RDGN is thus a paradigm of how evolution uses the finite genetic tools available to establish the structural and functional diversity observed in biology.

## Methods

### Fly stocks and transgenes used in this study

Drosophila stocks were reared at 25˚C on Nutrifly (Genesee Scientific) for the initial DGRP validation screen. Subsequent experiments were performed using standard yeast fly food recipe (See Table 1, below). Stocks were ordered from the Bloomington Drosophila Stock Center (BDSC), Vienna Drosophila RNAi Center (VDRC) or from the Zurich Orfeome Project (FlyORF). The following GAL4 lines were used: *Dilp2-GAL4R; btl-GAL80* (*BDSC# 37516*, described in [16]), *OK107-GAL4* [50] and *Dilp2-GAL4*[215-1-1-1] [40]. To visualize IPCs, *Dilp2-GAL4* lines controlled expression of *UAS-CD8GFP (BDSC# 5137)* or *UAS-nlsGFP (BDSC# 4776)*.

A list of the RNAi lines used in the DGRP screen are included in S3 Table.

RNAi lines for the RDGN follow-up experiments include *hth-RNAi (BDSC# 34637), ey-RNAi (BDSC# 32486), dac-RNAi(BDSC# 35022), exd-RNAi #1 (BDSC# 34897), tsh-RNAi (BDSC# 35030), optix-RNAi (BDSC# 55306), toy-RNAi (BDSC# 29346), toe-RNAi (BDSC# 50660), dan-RNAi (BDSC# 51501), eyg-RNAi (BDSC# 26226), gro-RNAi (BDSC# 35759), so-RNAi (BDSC# 35028)* and *eya-RNAi (BDSC# 35725)*. Overexpression lines include *UAS-Hth* (F000241) and *UAS-Exd* (F004513).

### Candidate gene identification

GWA analyses on *dilp* values were performed using the DGRP2 webtool (http://dgrp2.gnets.ncsu.edu/) [25,51]. Mean FPKM values of *dilp2*, *-3* and *-5* were used as the input and genes associated with variants that reached a nominal significance threshold of $p < 10^{-3}$ in the mixed effects model in one of the sexes were considered for further validation. Input files are provided in S1 Table.

### Transcriptome data

IPCs were fluorescently labelled with dilp2-GAL4; UAS-mCD8GFP. GFP-positive cells were manually sorted according to Nagoshi et al [52]. In short, 100 brains were dissected from adult flies, 3–5 days after eclosion into ice-cold dissecting solution (9.9 mM HEPES-KOH buffer, 137 mM NaCl, 5.4 mM KCl, 0.17 mM $NaH_2PO_4$, 0.22 mM $KH_2PO_4$, 3.3 mM glucose, 43.8 mM sucrose, pH 7.4) containing 50 μM d(−)-2-amino-5-phosphonovaleric acid (AP5), 20 μM 6,7-dinitroquinoxaline-2,3-dione (DNQX), 0.1 μM tetrodotoxin (TTX), and immediately transferred into modified SMactive medium (5 mM Bis-Tris, 50 μM AP5, 20 μM DNQX, 0.1 μM TTX) on ice. Brains were digested with l-cysteine-activated papain (50 units ml$^{-1}$ in dissecting saline) for 20 min at 25˚C. Digestion was quenched with a fivefold volume of the medium, and brains were washed twice with the chilled medium. Brains were triturated with a

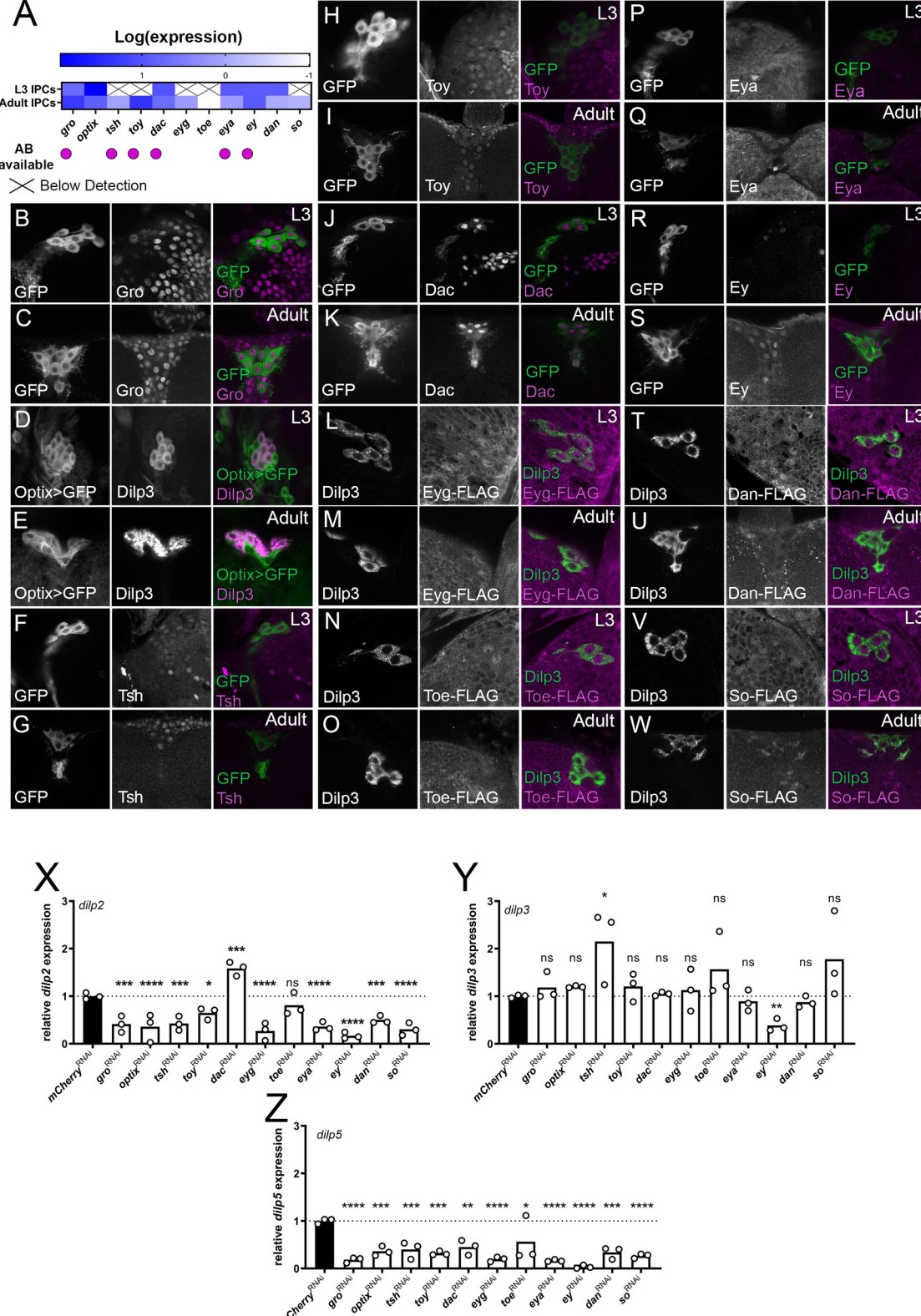

**Fig 7. Most RDGN transcription factors are expressed in the IPCs and are required for normal adult *dilp* expression.** (A). Additional RDGN transcription factors are expressed in larval and/or adult IPCs (Larval transcriptome data is taken from Cao et al., (2014) [39] while adult transcriptome data was generated for this study. (B-W) Gro, Tsh, Toy, Dac and Ey are readily detected in L3

and adult IPCs, only Eyg exhibited no detectable protein expression in larval or adult IPCs. Adult *Dilp2-GAL4$^R$ > CD8GFP* flies were reared to sexual maturity and brains were dissected and stained with available ABs directed against RDGN TFs. Expression of CD8GFP was used to label IPCs. For Optix, an Optix-GAL4 driver drove expression of the CD8GFP and was localized with an antibody against Dilp3, which labelled the IPCs. For Eyg, Toe, Dan and So, flies carrying a FLAG-GFP tagged version of these TFs were stained with an anti-FLAG antibody and anti-Dilp3 (X -Z) Genetic knockdown of all RDGN TFs in the IPCs using *Dilp2-GAL4$^{215-1-1-1}$* except *dac* and *toe* resulted in significant reduction of *dilp2* expression (one-way ANOVA with post-hoc Dunnett's test comparing to *mCherry$^{RNAi}$* controls, n = 3), *tsh* depletion in the IPCs increases dilp3 expression (one-way ANOVA with post-hoc Dunnett's test comparing to *mCherry$^{RNAi}$* controls, n = 3) and all were required for dilp5 expression (one-way ANOVA with post-hoc Dunnett's test comparing to *mCherry$^{RNAi}$* controls, n = 3). Flies were reared at controlled densities and RNA was collected from 10 female fly heads of flies controlled for age and diet.

flame-rounded 1,000-μl pipette tip with filter followed by a flame-rounded 200-μl pipette tip. Single, GFP-positive cells were selected, then reselected a further 2 times until approximately 100 fluorescent cells were pooled. Total RNA was extracted using a Nucleospin RNA micro kit (Macherey-Nagel). cDNA synthesis and adapter ligation were performed with the SMARTer cDNA synthesis kit (Takara Bio). 51bp paired-end reads were generated on an Illumina HiSeq2000 sequencer. Resulting reads were submitted to the European Nucleotide Archive, Accession Number PRJEB50165.

Quality assessment and read trimming was performed with fastp (v. 0.23.1) [53] with default settings. Reads were then pseudo-aligned to the reference Drosophila genome (BDGP6.32) and TPM values determined using salmon (v. 1.5.0) [54] and summarized to the gene level using tximport (v. 1.18.0) [55]

## RNA extraction and qPCR

For the validation of candidates from the DGRP, offspring was collected after eclosion and maintained at controlled densities of 20 total flies, aged to 3–5 days and snap frozen on dry ice. 10 to 15 fly heads were used per replicate for RNA extraction. For subsequent qPCR experiments concerning the RDGN transcription factors, flies were aged until 12–15 days old and only female flies were used for qPCR experiments. For larval qPCR, staged larvae (see below) were collected and all organs except the brain were dissected out. Each replicate is 5–6 larvae.

Total RNA was isolated using TRI reagent (Invitrogen) and reverse transcription was performed on 1μg RNA using Transcriptor First Strand Synthesis kit (Roche). Primer sequences used are listed in Table 2. qPCR was performed on a Step-One-Plus using the SYBR Green

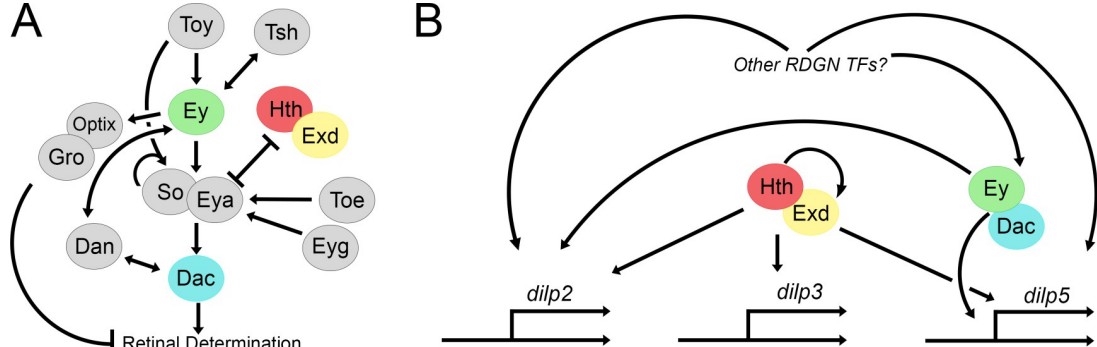

**Fig 8. A reshuffled RDGN regulates *dilp* expression in the IPCs.** (A) Schematic showing the regulatory relationships of the RDGN in the developing eye (modified from Kumar [33] and Bürgy-Roukala *et al.* [62]). (B) Speculative model of the RDGN in *dilp* regulation. Hth and Exd as a pair directly or indirectly regulate *dilp2, -3 and -5* expression, while Ey and Dac directly regulate *dilp5* epxression [32] and via other factors *dilp2* and *dilp3*. Other RDGN TFs directly or indirectly regulate *dilp2* and *dilp5 expression*.

**Table 1. Flyfood recipe.**

| Ingredient | Amount |
|---|---|
| Water | 15.2 L |
| Cornmeal | 1000 g |
| Yeast | 180 g |
| Agar | 112.5 g |
| Dextrose | 270 g |
| Molasses | 600 mL |
| Propionic Acid | 150 mL |
| 10%Hydroxybenzoate | 22.5 g |
| 99.8% Ethanol | 225 mL |

detection system. Transcript levels were normalized to Rp49 transcript levels (initial screen) or using the geometric means of RpS13 & Rp49 (transcription factor experiments). Mean ΔCt values were statistically compared as described below. Relative quantitation of transcript levels to control genotypes were calculated using the ΔΔCt method and plotted with GraphPad Prism software to visualize expression differences.

## Immunohistochemistry

Adult and larval brains were dissected in 1x PBS prior to 30 minute fixation with 4% formaldehyde. Primary antibodies were incubated overnight at 4°C in PAXD (PBS containing 1% BSA, 0.3% Triton X-100, 0.3% deoxycholate). Antibodies used in this study were mouse α-GFP (University of Iowa, Developmental Studies Hybridoma Bank (DSHB) 8H11; 1:100), rabbit α-GFP (Life Technologies, Ref#: A6455;

1:1000), mouse anti-FLAG (Sigma, F1804, 1:50), rabbit α-Hth (Dr. Richard Mann, 1:200), mouse α-Ey (DSHB E4H8, 1:5), rabbit α-Exd (Dr. Richard Mann, 1:200), mouse α-Exd (DSHB EXD B11M, 1:5), mouse α-Dac (DSHB mAbdac2-3, 1:20), α-Gro (DSHB anti-Gro, 1:10), rat α-Tsh (Dr. John Bell, 1:25), rat α-Toy (1:50), mouse α-Eya (DSHB eya10H6, 1:100), rat α-Dilp2 (Dr. Pierre Leopold, 1:1000), rabbit α-Dilp3 and rabbit α-Dilp5.

Rabbit antibodies directed against the Dilp3 partial peptide sequence DEVLRYCAAKPRT and against the Dilp5 peptide sequence RRDFRGVVDSCCRKS were generated as a service by Thermo Fischer Scientific Inc. For immunostaining, α-Dilp3 antibodies were used at a

**Table 2. Primers used in this study.**

| Gene | | Sequence (5' → 3') |
|---|---|---|
| *rps13* | Forward | GGG TCT GAA GCC CGA CAT T |
| | Reverse | GGC GAC GGC CTT CTTT GAT |
| *rp49* | Forward | GCC CAA GGG TAT CGA CAA CA |
| | Reverse | CTT GCG CTT CTT GGA GGA GA |
| rpl3 | Forward | AAG GAT GAC GCC AGC AAG CCA GTC |
| | Reverse | TAG CCG ACA GCA CCG ACC ACA ATC |
| *dilp2* | Forward | GCG AGG AGT ATA ATC CCG TGA T |
| | Reverse | GGA TTG AGG GCG TCC AGA |
| *dilp3* | Forward | GCA ATG ACC AAG AGA ACT TTG GA |
| | Reverse | GCA GGG AAC GGT CTT CGA |
| *dilp5* | Forward | GAG GCA CCT TGG GCC TAT TC |
| | Reverse | CAT GTG GTG AGA TTC GGA GCT |

dilution of 1:100 and α-Dilp5 antibodies used at a dilution of 1:400. Secondary antibodies used include goat α-mouse FITC & Cy3, goat α-rabbit FITC Cy3 & Cy5, goat α-rat Cy3 (Jackson Immunoresearch), donkey α-rabbit Alexa-594, donkey α-rat Alexa-594, donkey α-mouse Alexa-488 & Alexa-594. All secondary antibodies were used at a dilution of 1:200.

Samples were mounted in Vectashield Mounting Medium (Vector Laboratories Cat#H-1000). Immunohistochemistry images were taken with an Olympus FluoView FV1000 confocal microscope and processed using ImageJ64 (1.6.0_65; FIJI) [56,57] and Photoshop software.

## Oxidative stress resistance with Paraquat

Female flies were collected and aged until 12–15 days old, then transferred onto medium (1% agar, 10% Sucrose) containing 20mM Paraquat (Sigma 75365-73-0). Deaths were scored regularly and plotted in GraphPad Prism. Median lifespan was compared using a log-rank test with post-hoc Bonferonni correction for multiple testing.

## TAG determination

Groups of five 4–5 day old females flies, with ovaries and gut removed, were collected and total TAG was measured using the sulfo-phospho-vanillin method modified from [58]. In brief, flies were homogenised in chloroform:methanol (2:1) and incubated overnight at 37˚C with agitation. Samples were then placed in a 69˚C heatblock and chloroform:methanol was allowed to evaporate. Once fully evaporated, sulfuric acid (95–98%) is added and samples were placed on the heatblock for 10 minutes. Vanillin reagent (0.6% Vanillin, 34% phosphoric acid) is then added and samples are briefly mixed and incubated in the dark for 15 minutes. Samples are transferred in clear bottom plates and measured at 540 nm on a Synergy H1 spectrophotometer (BioTek). A standard curve made with Triolein was used to determine TAG content in samples.

## Developmental staging and time to eclosion

Parental genotypes (female *Dilp2-GAL4^R^;btlGAL80* and male *UAS-hth^RNAi^*, *UAS-exd^RNAi^*, or *UAS-mcherry^RNAi^*) were allowed to lay eggs on yeasted apple juice plates (1% agar, 10% Sucrose, 20% apple juice plates) for two to three hours. Embryos were allowed to develop to early L1 larvae at 25*C, collected and transferred to food vials at a defined larval density (20 larvae per vial). Larvae were then collected at specific timepoints for immunohistochemistry (68h AEL, 92h AEL and 100h AEL) or qPCR. (100h AEL).

Time until eclosion was scored by regularly assessing the number of eclosed flies. Time to eclosion was plotted in GraphPad Prism and average time until eclosion per genotype across vials (n = 4) was compared with a One-way ANOVA with post-hoc Dunnett's test.

## Determination of Adult Size Phenotypes

Adult flies 4–5 days after eclosion were collected from controlled conditions as described above, anesthetized using chloroform to prevent dessication, and weighed on a Mettler Toledo XS204 scale (Wet Weight; d = 0.1 mg). To measure wing size, a wing of chloroform anesthetized flies was removed and placed on a dissection slide. Wing length (L) was measured as a straight line from the alula opening to where the L3 vein meets the distal edge and wing width (W) was measured as a straight line starting from where the L5 vein meets the posterior edge to the anterior edge of the wing, perpendicular to the wing length line [59]. Total wing area (A) was then approximated by using the formula for an ellipse where $A = \pi * (L/2) * (W/2)$. Head width was measured by carefully removing heads from the thoraces of chloroform

anaesthetized flies and placed with its posterior side down on a slide. Head width is then measured as a straight line from the furthest point of the left eye from the midline to the furthest point on the right eye. Images of wings and heads were taken with an Olympus SZX12 stereo microscope and analyzed with Cell^D software

Mean weights between experimental and control flies were compared using a One-way ANOVA with post-hoc Dunnett's test with GraphPad software, and the relative change in adult weight between experimental and control genotypes were plotted.

### Female fecundity

Eclosed females were collected and allowed to mate for 48 hours with sibling males in a 4 female to 1 male sex ratio. Females were then collected and placed in groups of 4–5 in separate vials. Flies were transferred onto fresh food daily and the number of eggs was counted for 14 days. Mean number of eggs laid was compared with a One-way ANOVA with post-hoc Dunnett's test

### Nutritional challenges

For Hth and Exd nuclear localization experiments *Dilp2-GAL4$^R$;btlGAL80>CD8-GFP* flies were reared on normal food and female flies (24-48h) post eclosion were placed on three different diets (Starvation food: 1% agar, 1.2% glacial acetic acid; Sucrose only food: Starvation food supplemented with 15% sucrose; High protein: Starvation food supplemented with 40% yeast paste) for 72 hours followed by IHC. For nutritional dependency experiments *Dilp2-GAL4$^R$; btlGAL80>mcherry$^{RNAi}$, Dilp2-GAL4$^R$;btlGAL80>hth$^{RNAi}$, Dilp2-GAL4$^R$;btlGAL80>exd$^{RNAi}$* flies were placed on the three different nutritional conditions for 72h followed by qPCR.

### Statistics

All statistics were conducted using the GraphPad Prism software. All statistics were performed on raw data after normalization, where applicable (i.e. qPCR). For assays comparing means, unpaired T-tests for pairwise comparisons or one-way analysis of variance (ANOVA) tests for multiple comparisons were used. To test whether variance significantly differed between samples, an F-test was performed for pairwise comparisons and both Brown-Forsythe and Bartlett's Tests performed for multiple comparisons. If the standard deviations between samples were significantly different, a T-test with Welch's correction (Welch's T-test) was performed for multiple comparisons, whereas the Geisser-Greenhouse correction was always applied on all one-way ANOVA analyses, as sphericity of the data was not assumed. When comparing means to a single control sample, Dunnett's multiple comparisons test was used. For lifespan analysis under oxidative stress, median lifespan was compared with a log-rank test, with Bonferroni correction for multiple testing. For all statistical analyses, we assumed a significance level of 0.05.

### Hth motif identification

The Hth binding motif was identified in the JASPAR database [60] and FIMO [61] (Find Individual Motif Occurrences) from the MEME suite and was used to scan *dilp2*, *-3* and *-5* enhancers for putative Hth binding sites (p<0.001, both strands were scanned).

## Supporting information

**S1 Table. Input files for GWAS analyses.** Whole body expression data for *dilp2*, *-3*, and *-5* were collected from publicly available expression data of 183 DGRP lines.
(XLSX)

**S2 Table. Adult IPC transcriptome data.** Summary of RNAseq data of adult IPCs. Raw reads were submitted to the European Nucleotide Archive, accession number PRJEB50165.
(XLSX)

**S3 Table. List of 57 candidate genes.** 57 genes were selected for testing of their roles in IPCs to regulate *dilp2*, *-3*, and *-5* expression. The selection was based on the 125 genes identified through the GWAS analyses, comparison with adult IPC RNAseq results and availability of RNAi lines.
(XLSX)

**S1 Fig. Expressing an *mCherry^RNAi* in the IPCs does not affect *dilp* expression.** *dilp2*, *-3* and *-5* mRNA levels are not significantly different in *Dilp2-GAL4^R > mCherry^RNAi* compared to *Dilp2-GAL4^R /+* fly heads (unpaired two tailed Student's t-test $p > 0.05$).
(TIFF)

**S2 Fig. Hth knockdown in the IPCs reduces *Dilp2-GAL4* activity but identical effects are seen with second GAL4 line.** Representative images of adult female IPCs upon knockdown of mcherry (A) or Hth (B), IPCs are visualised by nuclear-GFP (nls-GFP) expression. Images were taken at identical confocal settings. (C) GFP fluorescence in the IPCs is significantly but mildly reduced upon hth depletion (Student's t-test, * $p<0.05$). Male (D) and female (E) *dilp2* expression is significantly reduced in *OK107-GAL4>hth^RNAi* flies compared to flies expressing an *mcherry^RNAi* control (Student's t-test, * $p<0.05$, **** $p<0.0001$). Adult weight is significantly reduced in *OK107-GAL4>hth^RNAi* male and female flies (Student's t-test, **** $p<0.0001$).
(TIFF)

**S3 Fig. Hth does not regulate Ey in in the IPCs.** Genetic depletion of *hth* in the IPCs does not affect Ey expression in the IPCs. IPCs are marked with CD8-GFP driven by *Dilp2-GAL4*.
(TIFF)

**S4 Fig. Hth expression in the IPCs is essential for nuclear localization of Exd in both larval and adult stages but not vice versa.** (A) Knockdown of *hth* results in the absence of Exd in larval IPC nuclei (arrow). (B) Knockdown of *exd* in larval IPCs does not affect Hth localization. (C) IPC-specific knockdown of *hth* results in the absence of Exd in IPC nuclei. (D) Hth localization is unaffected upon *exd* knockdown.
(TIFF)

**S5 Fig. Overexpression of Hth or Exd does not alter *dilp* expression or adult weight.** (A) Male or female flies overexpressing Hth or Exd in the IPCs do not weigh significantly more or less than control flies overexpressing GFP. (B) Overexpression of Hth or Exd does not alter *dilp2*, *-3 or -5* mRNA levels (ns, not significant, one-way ANOVA with post-hoc Dunnett's test to correct for multiple comparisons).
(TIFF)

**S6 Fig. Nuclear localization of Hth and Exd is not altered in different nutritional conditions and has nutrition-independent effects on *dilp* expression.** Nuclear localization of Hth and Exd in the IPCs is unaltered in starved (A), sucrose only (B) and protein rich (C) conditions. Flies were reared on control food and 24–48-hour old females were placed on different diets for 72 hours and subsequently dissected and stained. Female flies expressing *hth^RNAi* or *exd^RNAi* driven by *dilp2-GAL4^R;btlGAL80* were subjected to the same conditions as described above and collected for qPCR. Hth and Exd depletion had identical effects on *dilp2*, *-3* and *-5* expression levels in starved (E), sucrose only (F) and protein rich (G) as in control food (D).

(ns, not significant, * p < 0.05, ** p <0.01, *** p<0.001, **** p < 0.0001, Dunnett's multiple comparisons).
(TIFF)

**S7 Fig. Putative Hth binding sites within *dilp2*, *-3* and *-5* enhancers.** 2, 3 and 3 Hth motifs (**purple**) were identified in *dilp2*, *-3* and *-5* enhancers, respectively. Hth binding motif from the JASPAR database was used to scan in *dilp2*, *-3* and *-5* enhancers using the FIMO from the MEME suite (p<0.001, both strands). Start codon is highlighted in yellow.
(TIFF)

**S8 Fig. Hth but not Exd depletion affects Dilp2 protein levels in the IPCs in late L2 stage (68h after egg laying).** (A-B) Representative images of Dilp2 protein levels in *Dilp2-GAL4; btlGAL80> hth^RNAi* and *Dilp2-GAL4;btlGAL80> exd^RNAi* compared to control siblings (Cyo, left) reared in identical conditions. Flies were reared and age-controlled then brains from knockdown and balanced sibling flies were dissected and stained in the same tube. Expression of CD8GFP was used to differentiate control and test genotypes, and to label the IPCs. Images were acquired using identical confocal settings.
(TIFF)

**S9 Fig. Hth and Exd depletion reduces Dilp2, -3 and -5 protein levels in mid-L3 (92h AEL).** Representative images of Dilp2, 3 and -5 protein levels in (A-C) *Dilp2-GAL4;btlGAL80> hth^RNAi* and (D-F) *Dilp2-GAL4;btlGAL80> exd^RNAi* compared to control siblings (Cyo, left) reared in identical conditions. Flies were reared and age-controlled. Brains from knockdown and balanced sibling flies were dissected and stained in the same tube. Expression of CD8GFP was used to differentiate control and test genotypes, and label IPCs. Images were acquired using identical confocal settings.
(TIFF)

**S10 Fig. Knockdown of RDGN transcription factors does not affect IPC cell number.** Female flies expressing RNAis against RDGN transcription factors using Dilp2-GAL4;nls-GFP were dissected 4–6 days post-eclosion, fixed and mounted. The number of IPCs was determined by manually counting GFP+ nuclei. Numbers of brains analysed per genotype are shown inside the bars (ns, not significant, one-way ANOVA with post-hoc Dunnett's test to correct for multiple comparisons).
(TIFF)

## Author Contributions

**Conceptualization:** Kurt Buhler, Jason Clements, Sofie De Groef, Patrick Callaerts.

**Data curation:** Kurt Buhler.

**Formal analysis:** Mattias Winant, Kurt Buhler, Jason Clements, Sofie De Groef, Korneel Hens, Veerle Vulsteke, Patrick Callaerts.

**Funding acquisition:** Patrick Callaerts.

**Investigation:** Mattias Winant, Kurt Buhler, Jason Clements, Sofie De Groef, Korneel Hens, Veerle Vulsteke, Patrick Callaerts.

**Methodology:** Mattias Winant, Kurt Buhler, Jason Clements, Korneel Hens, Veerle Vulsteke, Patrick Callaerts.

**Project administration:** Patrick Callaerts.

**Resources:** Korneel Hens.

**Supervision:** Kurt Buhler, Jason Clements, Sofie De Groef, Patrick Callaerts.

**Validation:** Mattias Winant, Kurt Buhler, Jason Clements, Veerle Vulsteke.

**Visualization:** Mattias Winant.

**Writing – original draft:** Mattias Winant, Patrick Callaerts.

**Writing – review & editing:** Mattias Winant, Sofie De Groef, Korneel Hens, Patrick Callaerts.

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
