## [Decision Letter · Decision Letter 0]

5 Mar 2022

Dear Dr Callaerts,

Thank you very much for submitting your Research Article entitled 'Genome-wide analysis identifies Homothorax and Extradenticle as regulators of insulin in Drosophila Insulin-Producing cells' to PLOS Genetics.

The manuscript was fully evaluated at the editorial level and by independent peer reviewers. The reviewers appreciated the attention to an important problem, but raised some substantial concerns about the current manuscript. Based on the reviews, we will not be able to accept this version of the manuscript, but we would be willing to review a much-revised version. We cannot, of course, promise publication at that time.

If you decide to revise the manuscript for further consideration at PLOS Genetics, please aim to resubmit within the next 60 days, unless it will take extra time to address the concerns of the reviewers, in which case we would appreciate an expected resubmission date by email to plosgenetics@plos.org.

[LINK]

We are sorry that we cannot be more positive about your manuscript at this stage. Please do not hesitate to contact us if you have any concerns or questions.

Yours sincerely,

Ken M. Cadigan

Associate Editor

PLOS Genetics

Gregory P. Copenhaver

Editor-in-Chief

PLOS Genetics

Reviewer's Responses to Questions

**Comments to the Authors:**

Reviewer #1: This is an interesting study by Winant and colleagues analyzing on a broad level the inputs regulating dILP2,3,5 expression in Drosophila. The study identifies and focuses on Homothorax as a transcription factor regulating dILP expression. These dILPs are the fly equivalents of human insulin and insulin-like growth factor, and hence play an important role in fly development and physiology. Hence understanding their regulation is interesting and will be useful for a wide range of people working on fly development.

The main finding of this manuscript is the identification of Hth as a transcription factor regulating dILP expression. Essentially, Hth knockdown causes reduced dILP protein levels. It would be good to clarify if this is a direct or indirect regulation. Are there Hth binding sites near the dILP promoters or in an enhancer region? PMID 21483667 seems to have done a genome-wide ChIP for Hth - is there any useful information there? Can an enhancer be identified that is Hth responsive?

Minor Issues:

1. Fig 2B - it is hard to see, given such a good correlation, which genes are the dimorphic ones. It would be helpful if the dots in panel B could be labeled with the same colours as the ones used in panel B' depending on whether they are significant in one sex only, or in both sexes. (Presumably the ones further from the diagonal will be dimorphic - ie light grey?)

2. Fig 3 - reduced IPC immunostaining for the dILPs can either be due to reduced expression or increased secretion. Since Hth was identified in a Q-PCR screen for genes that affect dILP expression at the mRNA level, the phenotype seen in Fig 3 is most likely reduced expression. It would be helpful to the reader to show the Q-RT-PCR results as a graph in Fig 3 (similar to Fig 4C-E for exd) This cannot be appreciated so easily and qualitatively in the heat map in Fig 1C.

3. Why is there no effect of Hth knockdown in the IPCs on body weight? Shouldn’t such a dramatic drop in dILP2 expression cause a size effect?

4. Fig 3 - In dilp2-GAL4>Hth-RNAi animals, why does dILP2 expression drop so much while dILP2-GAL4 still works and drives expression of GFP? (ie why is there a discrepancy between expression of the endogenous dILP2 and the dILP2-GAL4?)

Reviewer #2: The manuscript by Winant and colleagues describes the interesting finding that the Homothorax-Extradenticle (Hth-Exd) complex regulates the expression of several insulin like peptides (ILPs) within the Insulin Producing Cells (IPCs) of Drosophila. The authors identified the Hth-Exd complex by analyzing the expression of Dilp genes within nearly 200 inbred fly lines. These lines are sequenced so gene variants can quickly be correlated with changes in expression. Of the 57 genes that the authors were able to interrogate through an RNAi screen, Hth and Exd had some of the strongest effects on Dilp expression. The authors go on to show that unlike the situation in the mammalian pancreas, the Hth-Exd complex does not regulate Pax6 transcription factors within the IPCs of Drosophila. The authors also show that several genes of the retinal determination network of which Hth-Exd and Pax6 are members, are expressed within the IPCs of the fly. And lastly, they show that depletion of the RD network members affects the expression of Dilps. A lot of work has gone into this manuscript and the data are quite convincing. However, in its present form, I am not sure that the results advance our understanding of Dilp regulation to a sufficient degree to warrant publication in PLoS Genetics. I have a few suggestions that I hope the authors will consider for a revised submission.

1. I think that a detailed phenotypic characterization of what happens to the fly when Hth and Exd are removed just from the IPCs will really strengthen the manuscript. As it stands now, the authors have identified an upstream regulator but have not shown us what happens when Hth-Exd are removed.

2. It would be helpful if the authors could correlate the regulation of Dilps by Hth-Exd to environmental conditions. Under which conditions is the Hth-Exd activated in the IPCs (ie. Starvation, well fed, constitutive?).

3. While it looks like the IPCs are present when the Hth-Exd complex is knocked down it would be helpful if the authors could provide a quantitative measure of how many IPCs are present in these knockdowns when compared to wild type. Changes in Dilp expression could be due to changes in IPC numbers – it will be helpful to see if the IPC numbers change or not. Similarly, I would suggest that the authors do the same for the other RD members. The results might strengthen their model or change it (which could be very interesting as well).

4. What happens to Dilp expression if the levels of the Hth-Exd complex are elevated within the IPCs.

**Have all data underlying the figures and results presented in the manuscript been provided?**

Reviewer #1: Yes

Reviewer #2: Yes

PLOS authors have the option to publish the peer review history of their article (what does this mean?). If published, this will include your full peer review and any attached files.

Reviewer #1: No

Reviewer #2: No

---

## [Decision Letter · Decision Letter 1]

15 Aug 2022

Dear Dr Callaerts,

We are pleased to inform you that your manuscript entitled "Genome-wide analysis identifies Homothorax and Extradenticle as regulators of insulin in Drosophila Insulin-Producing cells" has been editorially accepted for publication in PLOS Genetics. Congratulations!

Yours sincerely,

Ken M. Cadigan

Academic Editor

PLOS Genetics

Gregory P. Copenhaver

Editor-in-Chief

PLOS Genetics

Comments from the reviewers (if applicable):

Reviewer's Responses to Questions

**Comments to the Authors:**

Reviewer #1: The authors have addressed the issues raised in the original review.

Reviewer #2: The authors have answered my questions and I support the manuscript's publication in PLoS Genetics

**Have all data underlying the figures and results presented in the manuscript been provided?**

Reviewer #1: Yes

Reviewer #2: Yes

PLOS authors have the option to publish the peer review history of their article (what does this mean?). If published, this will include your full peer review and any attached files.

Reviewer #1: No

Reviewer #2: No

**Data Deposition**

http://datadryad.org/submit?journalID=pgenetics&manu=PGENETICS-D-22-00104R1

**Press Queries**

---

## [Editor Report · Acceptance letter]

7 Sep 2022

PGENETICS-D-22-00104R1 

Genome-wide analysis identifies Homothorax and Extradenticle as regulators of insulin in Drosophila Insulin-Producing cells 

Dear Dr Callaerts, 

We are pleased to inform you that your manuscript entitled "Genome-wide analysis identifies Homothorax and Extradenticle as regulators of insulin in Drosophila Insulin-Producing cells" has been formally accepted for publication in PLOS Genetics! Your manuscript is now with our production department and you will be notified of the publication date in due course.

With kind regards,

Agnes Pap

PLOS Genetics

On behalf of:
